# Beyond L1: Faster and Better Sparse Models with skglm

**Quentin Bertrand**
Mila & UdeM, Canada
quentin.bertrand@mila.quebec

**Quentin Klopfenstein**
Luxembourg Centre for Systems Biomedicine
University of Luxembourg
Esch-sur-Alzette, Luxembourg

**Pierre-Antoine Bannier**
Independent Researcher

**Gauthier Gidel**
Mila & UdeM, Canada
Canada CIFAR AI Chair

**Mathurin Massias**
Univ. Lyon, Inria, CNRS, ENS de Lyon,
UCB Lyon 1, LIP UMR 5668, F-69342
Lyon, France

## Abstract

We propose a new fast algorithm to estimate any sparse generalized linear model with convex or non-convex separable penalties. Our algorithm is able to solve problems with millions of samples and features in seconds, by relying on coordinate descent, working sets and Anderson acceleration. It handles previously unaddressed models, and is extensively shown to improve state-of-art algorithms. We release `skglm`, a flexible, `scikit-learn` compatible package, which easily handles customized datafits and penalties.

## 1 Introduction

Sparse generalized linear models play a central role in modern machine learning and signal processing. The Lasso (Tibshirani, 1996) and its derivatives (Zou and Hastie, 2005; Ng, 2004; Candes et al., 2008; Simon et al., 2013) have found numerous successful applications to large scale tasks in genomics (Ghosh and Chinnaiyan, 2005), vision (Mairal, 2010), or neurosciences (Strohmeier et al., 2016). This impact was made possible by two key factors: efficient algorithms and software implementations.

State-of-the-art algorithms for "smooth + non-smooth separable" problems predominantly rely on coordinate descent (CD, Tseng and S.Yun 2009; Nesterov 2012), which, when it can be applied, is more efficient than full gradient methods (Richtárik and Takáč, 2014, Sec. 6.1). Coordinate descent can even be improved with Nesterov-like acceleration, to obtain improved convergence rates (Lin et al., 2014; Fercoq and Richtárik, 2015). However, these better rates may fail to reflect in practical accelerations. On the contrary, Bertrand and Massias (2021) relied on Anderson acceleration (Anderson, 1965) to provide both better rates and practical acceleration for coordinate descent.

Even with efficient algorithms such as coordinate descent, the practical use of sparsity hits a computational barrier for problems with more than millions of features (Le Morvan and Vert, 2018). Multiple techniques have been proposed to make coordinate descent scale to huge problems. Notably, algorithms can be accelerated by reducing the number of variables to optimize over, using screening rules or working sets. Screening rules discard features from the problem in advance (El Ghaoui et al. 2010; Bonnefoy et al. 2015) or dynamically (Fercoq et al., 2015; Ndiaye et al., 2017). On the other side, working sets (Johnson and Guestrin, 2015; Massias et al., 2018) iteratively solve larger subproblems and progressively include variables identified as relevant.

For the Lasso and a few convex models, coordinate descent has been broadly disseminated to practitioners in off-the-shelf packages such as `glmnet` (Friedman et al., 2007) or `scikit-learn`

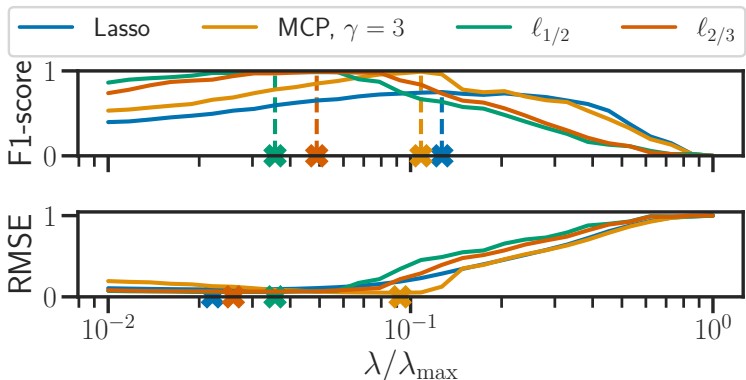

Figure 1: **Regularization paths computed with our algorithm.** Non-convex sparse penalties behave better than the L1 norm. Due to their lower bias, they achieve perfect support recovery, lower prediction error and their optimal regularization strength $\lambda$ in estimation (top) and prediction (bottom) correspond.

(Pedregosa et al., 2011). More recently, `celer`, a state-of-the-art convex working set algorithm (Massias et al., 2020) allowed for successful applications of the Lasso in large scale problems in medicine (Reidenbach et al., 2021; Kim et al., 2021) or seismology (Muir and Zhan, 2021).

Yet the Lasso is limited: non-convex sparse models enjoy better theoretical and empirical properties (Breheny and Huang, 2011; Soubies et al., 2015). As illustrated in Figure 1, they yield sparser solutions than convex penalties and mitigate the intrinsic Lasso bias. Yet, they have not so often been applied to huge scale applications. This is mostly an algorithmic barrier: while coordinate descent can be applied to non-convex penalties (Breheny and Huang, 2011; Mazumder et al., 2011; Bolte et al., 2014), screening rules and working sets are heavily dependent on convexity or quadratic datafits (Rakotomamonjy et al., 2019, 2022).

In this work, we solve this issue by designing a **state-of-the-art generic algorithm** to solve a wide range of sparse generalized linear models. The contributions are the following:

- We propose a non-convex converging working set algorithm relying on Anderson accelerated coordinate descent. For a specific class of non-convex penalties, we show:
  (a) Convergence of the proposed working set algorithm (Proposition 5).
  (b) Support identification of coordinate descent (Proposition 10).
  (c) Local convergence rates for the Anderson extrapolation (Proposition 13).
- We provide an extensive experimental comparison and we show state-of-the-art improvements on a wide range of convex and non-convex problems. In addition we release an efficient and modular python implementation, with a `scikit-learn` API, for practitioners to apply non-convex penalties to large scale problems.

## 2 Framework and proposed algorithm

### 2.1 Problem setting

In this paper, we consider problems of the form:

$$\hat{\beta} \in \underset{\beta \in \mathbb{R}^p}{\operatorname{argmin}} \ \Phi(\beta) \triangleq \underbrace{F(X\beta)}_{\triangleq f(\beta)} + \sum_{j=1}^{p} g_j(\beta_j) \ , \tag{1}$$

where $F$ is smooth, and the functions $g_j$ are proper and lower semicontinuous but not necessarily convex, whose proximal operator can be computed exactly. We write $g = \sum_j g_j$. Instances of Problem (1) include convex estimators: the Lasso, the elastic net, the sparse logistic regression, the dual of SVM with hinge loss. They also include non-convex penalties: $\ell_{0.5}$ and $\ell_{2/3}$ penalties (Foucart and Lai, 2009), the minimax concave penalty (MCP, Zhang 2010) or SCAD (Zhang, 2010), both with regression and classification losses. Formally, the assumptions are the following.

**Assumption 1.** *$f : \mathbb{R}^p \to \mathbb{R}$ is convex and differentiable and for all $j \in [p]$, the restriction of $\nabla_j f$ to the $j$-th coordinate is $L_j$-Lipschitz: for all $(x, h) \in \mathbb{R}^p \times \mathbb{R}$, $|\nabla_j f(x + he_j) - \nabla_j f(x)| \leq L_j |h|$.*

**Assumption 2.** *For any $j \in [p]$, $g_j : \mathbb{R} \to \mathbb{R}$ is proper, closed, and lower bounded.*

Following Attouch and Bolte 2009; Bolte et al. 2014 we focus on finding a critical point of $\Phi$.

**Definition 3.** *Using the Fréchet subdifferential (Kruger, 2003), a critical point $x \in \mathbb{R}^p$ is a point which satisfies $-\nabla f(x) \in \partial g(x)$.*

Assumptions 1 and 2 are usual, and, under boundedness of the iterates, ensure convergence of forward-backward and coordinate descent algorithms to a critical point (Attouch et al. 2013, Thm 5.1, Bolte et al. 2014, Thm. 3.1). In addition, our work focuses on the case where $g_j$'s present non-differentiability points, leading to the following extended notion of sparsity.

**Definition 4** (Generalized support). *The generalized support of $\beta \in \mathbb{R}^p$ is the set of indices $j \in [p]$ such that $g_j$ is differentiable at $\beta_j$: $\mathrm{gsupp}(\beta) = \{j \in [p] : \partial g_j(\beta_j) \text{ is a singleton}\}$.*

Penalties such as $\ell_1$, $\ell_q$ ($0 < q < 1$), MCP or SCAD are only not differentiable at 0, and this corresponds to the usual notion of sparsity. But Definition 4 goes beyond sparsity and extends to estimators such as SVM, where $g_j = \iota_{[0,C]}$ and the generalized support is the complement of the support vectors' set $\{j \in [p] : \beta_j = 0 \text{ or } \beta_j = C\}$. The generalized support of a critical point is usually of cardinality much smaller than $p$, and its knowledge makes the problem easier and faster to solve. Our working set algorithm exploits this structure in order to converge faster.

## 2.2 Proposed algorithm

The proposed algorithm exploits two main ideas:

- A working set strategy, able to handle a large class of convex and non-convex penalties (Algorithm 1).

- An Anderson accelerated coordinate descent for non-convex problems (Algorithm 2). The building blocks of Algorithm 2, coordinate descent (CD, Algorithm 3) and Anderson extrapolation (Anderson, Algorithm 4), can be found in Appendix A.

To avoid wasting computation on features outside the generalized support, working set algorithms iteratively select a subset of coordinates deemed important (the *working set*), and solve Problem (1) restricted to them. The key question is thus the notion of *important* features. Stemming from Definition 3, we rank features by their violation of the optimality condition: $\mathrm{score}_j^{\partial} = \mathrm{dist}(-\nabla_j f(\beta), \partial g_j(\beta))$ . For example, the MCP Fréchet subdifferential at 0 is $\partial g_j(0) = [-\lambda, \lambda]$, and the proposed score reads

$$\mathrm{score}_j^{\partial} = \begin{cases} \max\{0, |\nabla_j f(\beta)| - \lambda\} & \text{if} \quad \beta_j = 0 \ , \\ |\nabla_j f(\beta) + \nabla g_j(\beta_j)| & \text{otherwise} \ . \end{cases} \tag{2}$$

To control the working set growth, we use $\mathrm{score}_j^{\partial}$ to rank the features. Then, with $n_k = \max(n_{k-1}, 2 \, |\mathrm{gsupp}(\beta^{(t)})|)$ we take the $n_k$ largest of them in the working set, while retaining features currently in the working set. This growth quickly rises to the unknown size of the generalized support while avoiding overshooting, as backed up by recent theory in Ndiaye and Takeuchi (2021).

**Proposition 5.** *Let $\mathcal{W}_t$ be the $t$-th working set. Suppose that Algorithm 2 converges toward a critical point, and for all $t \geq 0$, $\mathcal{W}_t \subset \mathcal{W}_{t+1}$, then the iterates of Algorithm 1 converge towards a critical point of Problem (1).*

**Algorithm 1** `skglm` (proposed)

**input :** $X, \beta \in \mathbb{R}^p, n_{\text{out}} \in \mathbb{N},$
$\quad\quad\quad n_{\text{in}} \in \mathbb{N}, \text{ws\_size} \in \mathbb{N}, \epsilon > 0$

1   **for** $t = 1, \dots, n_{\text{out}}$ **do**
2      $\text{score} = \big( \text{dist}\, (-\nabla_j f(\beta), \partial g_j(\beta_j)) \big)_{j \in [p]}$
3      $\text{ws\_size} = \max(\text{ws\_size}, 2 \times |\text{gsupp}(\beta)|)$
      `// ws_size features with largest`
        `scores`
4      $\text{ws} = \text{arg\_topK}(\text{score}, K = \text{ws\_size})$
5      **if** $\max_{j \in [p]} \text{dist}\, (-\nabla_j f(\beta), \partial g_j(\beta_j)) \leq \epsilon$
      **then** stop
6      **else** `// accelerated CD on working set`
7      $\beta \leftarrow \text{inner\_solver}(X, \beta, \text{ws}, n_{\text{in}}, \epsilon)$
8   **return** $\beta$

---

**Algorithm 2** `inner_solver`

**input :** $X, \beta^{(0)} \in \mathbb{R}^p, \text{ws} \subset [p], n_{\text{in}}, \epsilon, M = 5$

1   **for** $k = 1, \dots, n_{\text{in}}$ **do**
2      $\beta^{(k)} \leftarrow \text{CD}(X, \beta^{(k-1)}, X\beta, \text{ws})$ `// Algo. (3)`
3      **if** $k \bmod M = 0$ **then**
       `// Algo. (4), ` $\mathcal{O}(M^2|\text{ws}| + M^3)$
4        $\beta_{\text{ws}}^{\text{extr}} \leftarrow \text{Anderson}(\beta_{\text{ws}}^{(k-M)}, \dots, \beta_{\text{ws}}^{(M)})$
       `// test objective ` $\mathcal{O}(n|\text{ws}|)$
5        **if** $\Phi(\beta_{\text{ws}}^{\text{extr}}) < \Phi(\beta_{\text{ws}}^{(k)})$ **then**
6          $\beta_{\text{ws}}^{(k)} \leftarrow \beta_{\text{ws}}^{\text{extr}}; X\beta \leftarrow X_{\text{ws}}\beta_{\text{ws}}^{\text{extr}}$
7      **if** $\max_{j \in \text{ws}} \text{dist}\, (-\nabla_j f(\beta), \partial g_j(\beta_j)) \leq \epsilon$
      **then** stop
8   **return** $\beta^{(k)}$

---

Proof of Proposition 5 can be found in Appendix B.1. The second key ingredient to our algorithm is to use state-of-the-art Anderson accelerated coordinate descent for non-convex problems. In Section 2.3 we show that coordinate descent yields finite time support identification for a large class of non-convex problems (Proposition 10), which leads to acceleration (Proposition 13). As experiments demonstrate in Section 3, this rate allows our algorithm to surpass state-of-the-art solvers.

### 2.3   Anderson accelerated coordinate descent analysis for $\alpha$-semi-convex penalties

We now turn to our main technical contributions: we show that Algorithm 2 achieves finite time support identification (Proposition 10) of the generalized support (Definition 4) for specific class of non-smooth non-convex penalties (Assumption 6), which includes the MCP (Proposition 7). Based on Proposition 10, we are able to derive convergence rates for Anderson acceleration (Proposition 13).

We study our inner solver (Algorithm 2); for convenience we still refer to $\beta$ and $X$ for their counterparts restricted to the working set. The following assumptions are required.

**Assumption 6** ($\alpha$-semi-convex)**.** *For all $j \in [p]$ $g_j/L_j$ is $\alpha$-semi-convex, i.e., $g_j/L_j + \alpha\|\cdot\|^2/2$ is convex, with $\alpha < 1$.*

Note that in statistics, the admissible value range of hyperparameters for MCP and SCAD are datafit-dependent, (see Breheny and Huang 2011, Sec. 2.1, normalized columns and $\gamma > 1 = 1/\|X_{:j}\| = 1/L_j$ or Soubies et al. 2015, Eq. 4.2) and yields $\alpha$-semi-convexity for MCP and SCAD[1].

**Proposition 7** ($\alpha$-semi-convexity of MCP)**.** *Let* $\text{MCP}_{\lambda,\gamma}(x) \triangleq \begin{cases} \lambda|x| - \frac{x^2}{2\gamma} , & \text{if } |x| \leq \gamma\lambda , \\ \frac{1}{2}\gamma\lambda^2 , & \text{if } |x| > \gamma\lambda . \end{cases}$

*If $\gamma > 1/L_j$, then $\text{MCP}_{\lambda,\gamma}/L_j$ is $\alpha$-semi-convex with $\alpha = \frac{1}{2}(1 + \frac{1}{\gamma L_j})$ (i.e., Assumption 6 holds).*

Note that Assumption 6 does not hold for the $\ell_q$-penalties ($0 < q < 1$), for which we propose an alternative in Appendix C.

**Assumption 8** (Existence)**.** *Problem (1) admits at least one critical point.*

In Proposition 10, convergence of Algorithm 2 toward a critical point $\hat\beta$ is assumed, and the following assumption is made on this critical point.

**Assumption 9** (Non degeneracy)**.** *The considered critical point $\hat\beta \in \mathbb{R}^p$ is non-degenerated: for all $j \notin \text{gsupp}(\hat\beta)$,*

$$-\nabla f_j(\hat\beta) \in \text{interior}(\partial g_j(\hat\beta_j)). \tag{3}$$

Assumption 9 is a generalization of qualification constraints (Hare and Lewis, 2007, Sec. 1), and is usual in the machine learning literature (Zhao and Yu, 2006; Bach, 2008; Vaiter et al., 2015). For

---

[1]However MCP and SCAD are not $\alpha$-semiconvex for all hyperparameter values.

the $\ell_1$-norm, if the entries of the design matrix $X$ are drawn from an i.i.d normal distribution, then Assumption 9 holds with high probability (Candes and Tao, 2005; Rudelson and Vershynin, 2008).

Equipped with the previous assumptions we show that coordinate descent achieves model identification for this class of non-convex problems.

**Proposition 10** (Model identification of CD). *Suppose*

1. *Assumptions 1, 2, 6 and 8 hold.*
2. *The sequence $(\beta^{(k)})_{k \geq 0}$ generated by coordinate descent (Algorithm 2 without extrapolation) converges toward a critical point $\hat{\beta}$.*
3. *Assumption 9 holds for $\hat{\beta}$.*

*Then, Algorithm 2 (without extrapolation) identifies the model in finitely many iterations: there exists $K > 0$ such that for all $k \geq K$, $\beta_{\mathcal{S}^c}^{(k)} = \hat{\beta}_{\mathcal{S}^c}$.*

In other words, for $k$ large enough, $\beta^{(k)}$ shares the generalized support of $\hat{\beta}$. The identification property was proved for a proximal gradient descent algorithm in the non-convex case (Liang et al., 2016) under the assumption that the non-smooth function $g$ is partly smooth (Lewis, 2002). For ourselves, Proposition 10 not rely on the partly smooth assumption to ensure identification property. Authors are not aware of previous identification results for coordinate descent in the non-convex case.

In addition, if $f$ and $g$ are locally regular on the generalized support at the considered critical point, our algorithm enjoys local acceleration when combined with Anderson extrapolation (Proposition 13).

**Assumption 11** (Locally $\mathcal{C}^3$). *For all $j \in \mathcal{S} \triangleq \mathrm{gsupp}(\hat{\beta})$, $g_j$ is locally $\mathcal{C}^3$ around $\hat{\beta}_j$, and $f$ is locally $\mathcal{C}^3$ around $\hat{\beta}$.*

Assumption 11 on the function $f$ is mild and holds for usual machine learning datafitting terms. Assumption 11 on the functions $g_j$, $j \in \mathcal{S}$, is stronger: for instance, for the MCP, it implies $\hat{\beta}_j \neq \gamma\lambda$ for all $j \in \mathcal{S}$. However this assumption is standard in the literature, see Liang et al. 2016, Sec. 3.3

**Assumption 12.** *(Local strong convexity) The Hessian of $f$ at the considered critical point $\hat{\beta} \in \mathbb{R}^p$, restricted to its generalized support $\mathcal{S}$, is positive definite, i.e., $\nabla_{\mathcal{S},\mathcal{S}}^2 f(\hat{\beta}) + \nabla_{\mathcal{S},\mathcal{S}}^2 g(\hat{\beta}) \succ 0$.*

Assumption 12 requires local strong convexity restricted to the generalized support $\mathcal{S}$, which is standard in the MCP / SCAD literature (Breheny and Huang 2011, Section 4.1) and is usual to derive local linear rates of convergence (Liang et al., 2016, Section 3.3). For instance, for the Lasso, if the entries of the design matrix $X$ are drawn from a continuous distribution, then Assumption 12 holds with probability one (Tibshirani, 2013, Lemma 4).

**Proposition 13.** *Consider a critical point $\hat{\beta}$ and suppose*

1. *Assumptions 1, 2 and 8 hold.*
2. *The functions $f$ and $g_j$, $j \in [p]$ are piecewise quadratic (which is the case for the MCP regression).*
3. *The sequence $(\beta^{(k)})_{k \geq 0}$ generated by Anderson accelerated coordinate descent with updates from 1 to $p$ and $p$ to 1 (Algorithm 2 with extrapolation) converges to a critical point $\hat{\beta}$.*
4. *Assumptions 9, 11 and 12 hold for $\hat{\beta}$.*

*Then there exists $K \in \mathbb{N}$, and a $\mathcal{C}^1$ function $\psi : \mathbb{R}^{|\mathcal{S}|} \to \mathbb{R}^{|\mathcal{S}|}$ such that, for all $k \in \mathbb{N}, k \geq K$:*

$$\beta_j^{(k)} = \hat{\beta}_j \text{, for all } j \in \mathcal{S}^c, \tag{4}$$

*Let $T \triangleq \mathcal{J}\psi(\hat{\beta})$, $H \triangleq \nabla_{\mathcal{S},\mathcal{S}}^2 f(\hat{\beta}) + \nabla_{\mathcal{S},\mathcal{S}}^2 g(\hat{\beta})$, $\zeta \triangleq (1 - \sqrt{1 - \rho(T)})/(1 + \sqrt{1 - \rho(T)})$ and $B \triangleq (T - \mathrm{Id})^\top (T - \mathrm{Id})$. Then $\rho(T) < 1$ and the iterates of Anderson extrapolation enjoy local accelerated convergence rate:*

$$\|\beta_{\mathcal{S}}^{(k-K)} - \hat{\beta}_{\mathcal{S}}\|_B \leq \left(\sqrt{\kappa(H)}\frac{2\zeta^{M-1}}{1+\zeta^{2(M-1)}}\right)^{(k-K)/M} \|\beta_{\mathcal{S}}^{(K)} - \hat{\beta}_{\mathcal{S}}\|_B . \tag{5}$$

The proof can be found in Appendix B.5.

Table 1: Most popular packages for sparse generalized linear models.

| Name | Acceleration | Huge scale | Nncvx | Modular |
|---|---|---|---|---|
| glmnet (Friedman et al., 2010) | ✗ | ✗ | ✗ | ✗ (Fortran) |
| scikit-learn (Pedregosa et al., 2011) | ✗ | ✗ | ✗ | ✗ (Cython) |
| lightning (Blondel and Pedregosa, 2016) | ✗ | ✗ | ✗ | ✓ (Cython) |
| celer (Massias et al., 2018) | ✓ | ✓ | ✗ | ✗ (Cython) |
| picasso (Ge et al., 2019) | ✗ | ✗ | ✓ | ✗ (C++) |
| pyGLMnet (Jas et al., 2020) | ✗ | ✗✗ | ✗ | ✓ (Python) |
| fireworks (Rakotomamonjy et al., 2022) | ✗ | ✓ | ✓ | N.A. (Python) |
| skglm (ours) | ✓ | ✓ | ✓✓ | ✓ (Python) |

**Related work.** Most Anderson acceleration convergence results are shown for quadratic objectives for specific algorithms: gradient descent (Golub and Varga, 1961; Anderson, 1965), ADMM (Poon and Liang, 2019), coordinate descent (Bertrand et al., 2020). Outside of the quadratic case, convergence results are usually significantly weaker (Scieur et al., 2016; Sidi, 2017; Brezinski et al., 2018; Mai and Johansson, 2019; Ouyang et al., 2020). Regarding the smooth non-convex case, Wei et al. (2021) proposed a stochastic Anderson acceleration and proved convergence towards a critical point. Proposition 13 generalizes Scieur et al. (2020, Prop 2.1) and Bertrand and Massias (2021, Prop. 4) to the proximal convex and $\alpha$-semi-convex cases. To our knowledge this is one of the first quantitative results for Anderson acceleration in a non-convex setting.

## 2.4 Comparison with existing work

In this section we compare our contribution to existing algorithms and implementations, which are summarized in Table 1. *Huge scale* refers to the fact that the algorithm can run on problems with millions of variables. *Non-convex* tells if the algorithm handles non-convex penalties. *Modular* indicates that it is easy to add a new model, through a different datafitting term or penalty.

The packages glmnet (Friedman et al., 2010), scikit-learn (Pedregosa et al., 2011) and lightning (Blondel and Pedregosa, 2016) implement coordinate descent (cyclic or random). They rely on compiled code such as Fortran or Cython, making it very difficult to implement new models[2] or faster algorithms like working set[3]. They do not handle non-convex penalties.

More recent algorithms such as blitz (Johnson and Guestrin, 2015), celer (Massias et al., 2018), picasso (Ge et al., 2019) or fireworks (Rakotomamonjy et al., 2022) use working set strategies. celer and blitz are state-of-the-art algorithms for the Lasso, but their score to prioritize features relies on duality. fireworks extends blitz to some non-convex penalties (writing as difference of convex functions), with $\text{score}_j^{\text{fireworks}} = \text{dist}(-\nabla_j f(\beta), \partial g_j(0))$. Yet this rule does not consider the subdifferential of $g$ at the current point, but at 0, which is a coarse information. Finally, fireworks, building upon the seminal non convex working set solver of Boisbunon et al. (2014), does not provide accelerated convergence rates and does not come with a public implementation. picasso (Ge et al., 2019) lacks modularity (penalties are hardcoded), and the solver is not suited for huge scale (it does not support large sparse matrices). Deng and Lan (2019) proposed an algorithm based on inertially accelerated coordinate descent, which fails to provide practical speedups according to Bertrand and Massias (2021).

Contrary to these algorithms, ours is generic and relies only on the knowledge of $\nabla f$ and $\text{prox}_g$. For any new penalty, this information can be written in a few lines of Python code, compiled with numba (Lam et al., 2015) for speed efficiency. We therefore improve state-of-the-art algorithms in the convex case, and generalize to virtually any datafit and penalty, even nonconvex.

## 3 Experiments

Our package relying on numpy and numba (Lam et al., 2015; Harris et al., 2020) is attached in the supplementary material. An open source, fully tested and documented version of the code can be

---

[2] https://github.com/scikit-learn/scikit-learn/pull/10745 (4 years old)
[3] https://github.com/scikit-learn/scikit-learn/pull/7853 (5 years old)

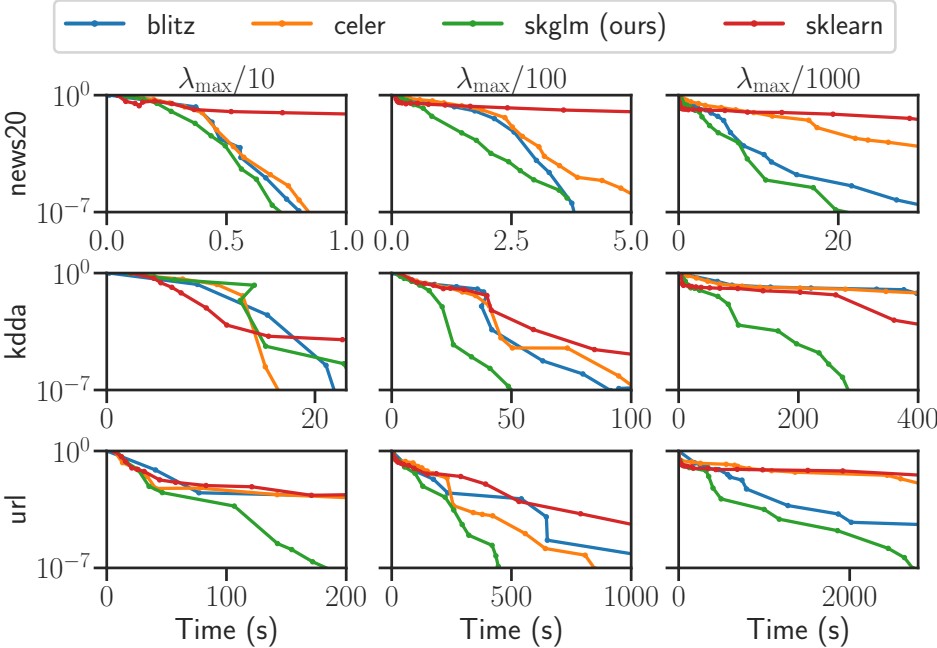

Figure 2: **Lasso, duality gap.** Normalized duality gap as a function of time for the Lasso on multiple datasets, for multiple values of $\lambda$.

found at https://github.com/scikit-learn-contrib/skglm. We use datasets from libsvm[4] (Fan et al. 2008, see table 2).

We compare multiple algorithms to solve popular Machine Learning and inverse problems: Lasso, Elastic net, multitask sparse regression, MCP regression. The compared algorithms are the following:

- scikit-learn (Pedregosa et al., 2011), which implements coordinate descent in Cython,
- celer (Massias et al., 2020), which combines working sets, screening rules, coordinate descent, and Anderson acceleration in the dual, in Cython,
- blitz (Johnson and Guestrin, 2015), which combines working sets with prox-Newton iterations (Lee et al., 2012) in C++,
- coordinate descent (CD, Tseng and S.Yun 2009),
- skglm (Algorithm 1, ours), using $M = 5$ iterates for the Anderson extrapolation.

**Other solvers.** Experiment per experiment, there exist niche solvers (such as aggressive Gap Safe Rules, Ndiaye et al. 2020). Since our goal is a *general purpose* algorithm able to deal with many models, we do not include them in the comparison. In addition, we focus on solving a single instance of Problem (1), rather than a regularization path (*i.e.,* a sequence of problems for multiple regularization strengths). As glmnet is designed to compute regularization paths, we could not include it in the comparison. The reader can refer to Johnson and Guestrin (2015, Fig. 4) or Figure 8 in Appendix E for comparisons on single optimization problems with glmnet; glmnet and additional algorithms are discussed in Appendix E.

**How to do a fair comparison between solvers?** To plot the convergence curves, we use the benchopt[5] benchmarking package (Moreau et al., 2022). In order to automate and reproduce optimization benchmarks it treats solvers as black boxes. It launches them several times with increasing maximum number of iterations, and stores the resulting objective values and times to reach it. As each point on a solver curve is obtained in a different run, the curves are not monotonic, and there may be several points corresponding to the same time. This merely reflects the variability in solvers running time across runs; we refer to Figure 10 in Appendix E.6 for the inevitability of this phenomenon with black box solvers.

[4] https://www.csie.ntu.edu.tw/~cjlin/libsvmtools/datasets/
[5] https://github.com/benchopt/benchopt

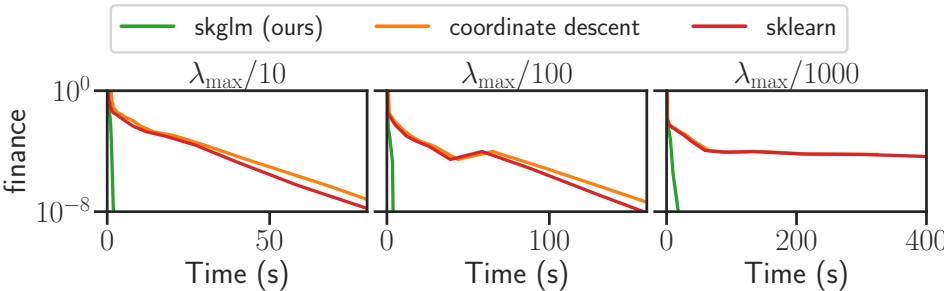

Figure 3: **Elastic net, duality gap.** Normalized duality gap as a function of time for the elastic net for multiple values of $\lambda$, $\rho = 0.5$.

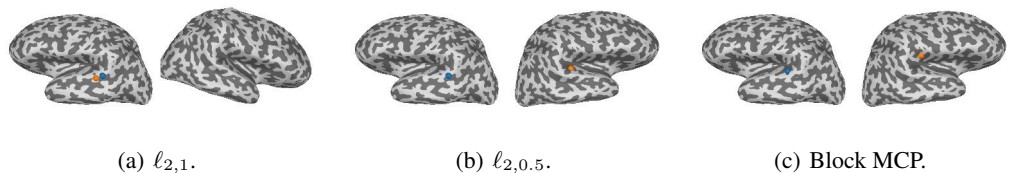

(a) $\ell_{2,1}$.        (b) $\ell_{2,0.5}$.        (c) Block MCP.

Figure 4: **Real data, brain source locations recovered by convex and non-convex penalties after a right auditory stimulation.** 4(a) shows that a convex penalty fails at identifying one source in each hemisphere, while 4(b) and 4(c) demonstrates the capability of non-convex penalties to recover the correct solution.

## 3.1 Convex problems

**Lasso.** In Figure 2 we compare solvers for the Lasso: ($f = \frac{1}{2n}\|y - X\cdot\|^2$, $g_j = \lambda|\cdot|$). We parametrize $\lambda$ as a fraction of $\lambda_{\max} = \|X^\top y\|_\infty/n$, smallest regularization strength for which $\hat\beta = 0$. For large scale datasets (*rcv1*, *news20*), `skglm` yields performances better or similar to the state-of-the-art algorithms `blitz` and `celer`. For huge scale datasets (*kdda* and *url*), `skglm` yields significant speedups over them. The improvement over the popular `scikit-learn` can be of two orders of magnitude. Thus, *while dealing with many more models, our algorithm still yields state-of-the-art speed for basic ones*.

**Elastic net.** Our approach easily generalizes to other problems, such as the elastic net ($f = \frac{1}{2n}\|y - X\cdot\|^2$, $g_j = \lambda(\rho|\cdot| + \frac{1-\rho}{2}(\cdot)^2)$). Figure 3 shows the duality gap as a function of time for `skglm` (ours), `sklearn`, and our numba implementation of coordinate descent. The proposed algorithm is orders of magnitude faster than `scikit-learn` and vanilla coordinate descent, in particular for large datasets and low regularization parameter values (*finance*, $\lambda_{\max}/1000$). Note that `blitz` does not implement a solver for the elastic net. Many Lasso solvers would easily handle the elastic net, but relying on Cython/C++ code makes the implementation time-consuming. By contrast, it takes 40 lines of code to define an $\ell_1 + \ell_2$-squared penalty with our implementation. An additional experiment on the dual of SVM with hinge loss is in Appendix E.4.

## 3.2 Non-convex problems

In this subsection we propose a comparison on two non convex problems.

**MCP regression.** MCP regression is Problem (1) with $f = \frac{1}{2n}\|y - X\cdot\|^2$, $g_j = \text{MCP}_{\lambda,\gamma}$ for $\gamma > 1$. As usual for this problem, we scale the columns of $X$ to have norm $\sqrt{n}$. On Figure 5, we compare our algorithm to `picasso` on a dense dataset ($n = 1000, p = 5000$); as this package does not support large sparse design matrices, for the *rcv1* dataset we use an iterative reweighted L1 algorithm (Candes et al., 2008). Since the derivative of the MCP vanishes for values bigger than $\lambda\gamma$, this approach requires solving weighted Lassos with some 0 weights. Up to our knowledge, our

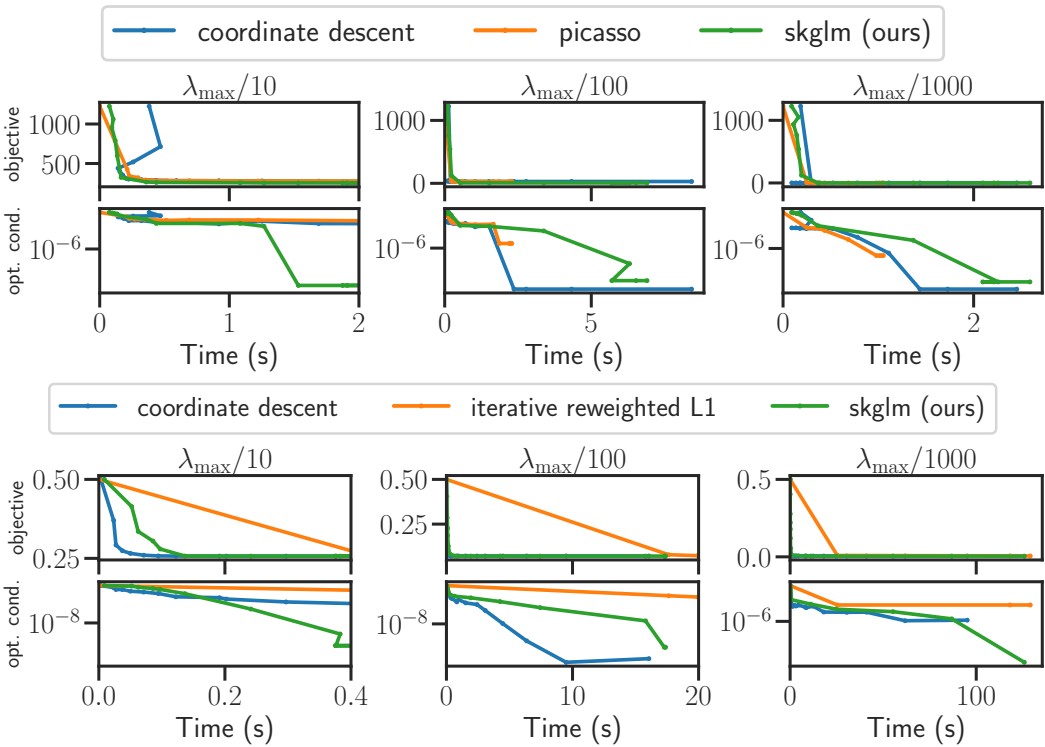

Figure 5: **MCP, objective value and violation of first order condition.** Objective value and violation of optimality condition of the iterates, $\text{dist}(-\nabla f(\beta^{(k)}), \partial g(\beta^{(k)}))$, as a function of time for the MCP for multiple values of $\lambda$ ($\gamma = 3$) on a simulated dense dataset (top) and the rcv1 dataset (normalized columns).

algorithm is the only efficient one with such a property. Our algorithm handles problems of large size, converges to a critical point, and, due to its progressive inclusion of features, is able to reach a sparser critical point than it competitors.

**Application to neuroscience** To demonstrate the usefulness of our algorithm for practitioners, we apply it to the magneto-/electroencephalographic (M/EEG) inverse problem. It consists in reconstructing the spatial cortical current density at the origin of M/EEG measurements made at the surface of the scalp. Non-convex penalties (Strohmeier et al., 2015) exhibit several advantages over convex ones (Gramfort et al., 2013): they yield sparser physiologically-plausible solutions and mitigate the $\ell_1$ amplitude bias. Here the setting is multitask: $Y \in \mathbb{R}^{n \times T}$ and thus we use block penalties (details in Appendix D). We use real data from the mne software (Gramfort et al., 2014); the experiment is a right auditory stimulation, with two expected neural sources to recover in each auditory cortex. In Figure 4, while the $\ell_{2,1}$ penalty fails at localizing one source in each hemisphere, the non-convex penalties recover the correct locations. This emphasizes on the critical need for fast solvers for non-convex sparse penalties as well as our algorithm's ability to handle the latter. In this work we focused on optimization-based estimators to solve the inverse problem, note that one could have resort to other techniques, such as Bayesian techniques (Ghosh and Doshi-Velez, 2017; Fang et al., 2020).

**Ablation study.** To evaluate the influence of the two components of Algorithm 1, an ablation study (Figure 6) is performed. Four algorithms are compared: with/without working sets and with/without Anderson acceleration. Figure 6 represents the duality gap of the Lasso as a function of time for multiple datasets and values of the regularization parameters $\lambda$ (parametrized as a fraction of $\lambda_{\max}$). First, Figure 6 shows that working sets always bring significant speedups. Then, when combined with working set, Anderson acceleration bring significant speed-ups, especially for hard problems with low regularization parameters. An interesting observation is that on large scale datasets (*news20* and

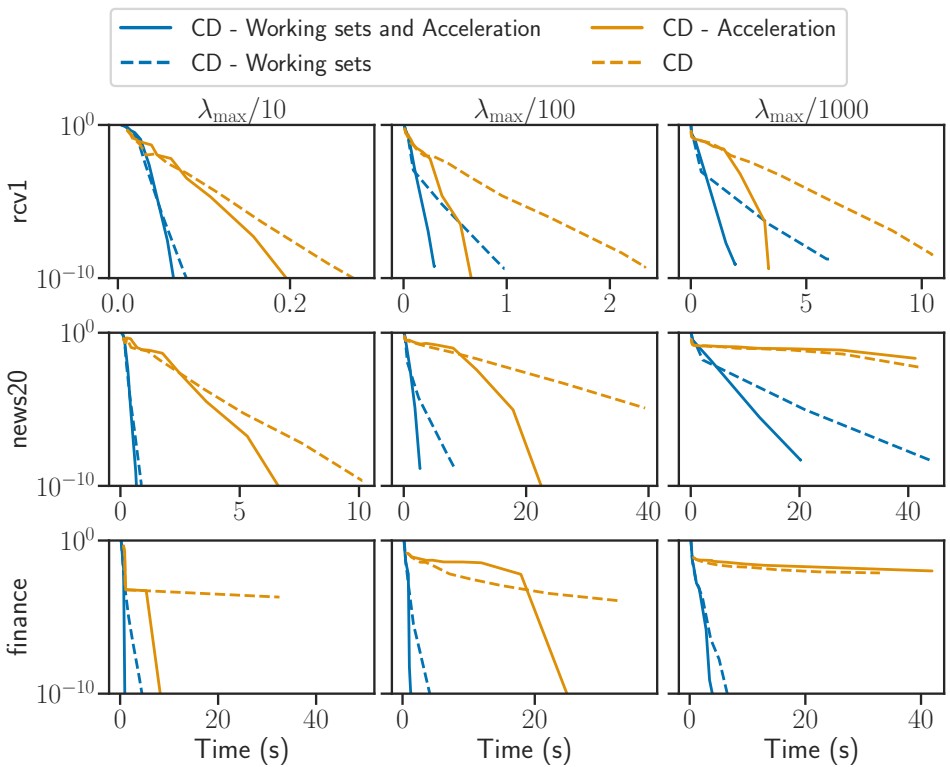

Figure 6: **Lasso, duality gap.** Normalized duality gap as a function of time for the Lasso.

*finance*) and for low regularization parameters ($\lambda_{max}/100$ and $\lambda_{max}/1000$) Anderson acceleration *without* working set does not bring acceleration. This highlights the importance of combining Anderson acceleration with working sets.

**Conclusion and broader impact.** In this paper, we have proposed an accelerated versatile algorithm for a specific class of non-smooth non-convex problems. Based on working sets, coordinate descent and Anderson acceleration, we have improved state of the art on convex problems, and handled previously out-of-reach problems. Thorough experiments demonstrated the speed and interest of our approach. A limitation of this work is the considered function class ($\alpha$-semi-convex), which can be seen as restrictive. One possible extension would be weakly convex functions (Davis and Drusvyatskiy, 2019, Sec. 1). We deeply believe that the high quality code provided will benefit to practitioners, and ease the use of non-convex penalties for real world problems, from neuroimaging to genomics. We proposed an optimization algorithm and do not see potential negative societal impacts.

## Acknowledgements

The experiments were ran on the CBP cluster of ENS de Lyon (Quemener and Corvellec, 2013). QB would like to thank Samsung Electronics Co., Ldt. for funding this research. GG is supported by an IVADO grant.

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
