# OpenReview forum: "Beyond L1: Faster and Better Sparse Models with skglm"
_NeurIPS.cc/2022/Conference — NeurIPS 2022 Accept_

### Official Review · Reviewer_783Y · 2022-07-07

**Rating:** 5
**Confidence:** 2
**Soundness:** 3 good
**Presentation:** 3 good
**Contribution:** 3 good

**Summary:**

The paper proposes a fast co-ordinate descent algorithm for sparse linear models with \alpha-semi-convex penalties. The algorithm includes two key step, one is to introduce a score to rank the variables and to get a working set; the other is to user Anderson acceleration in the inner-loop. The authors give a series of theoretical guarantees. The numerical experiments show the effectiveness of the proposed algorithm.

**Questions:**

See above

**Ethics Review Area:**

["I don’t know"]

**Strengths And Weaknesses:**

I am not an expert of optimization theory and I might not be authoritative enough to evaluate the theoretical contributions of the paper.

Pros:
1. the theoretical results seem rich and solid.

Cons:
1.  the algorithmic contributions are not clearly articulated. It seems working set and Anderson acceleration are known ideas, right? How novel is it to apply them in the co-ordinate descent framework? Is there any other CD algorithm that uses the two ideas? Also, is the proposed score function brand-new? The authors should clarify their contributions in the two critical steps.
2. The paper does not introduce the details about Anderson acceleration, not even including a brief summary.
3. Line 93, why n_k is bounded by 2 | gsupp(\beta^t)|? The authors do not explain the motivation and intuition. Can we change the factor 2? Since the size of the working set is important for the efficiency, the authors should give a clarification about the size choice.
4. The authors should be aware of Bayesian sparse learning works, such as based on the spike-and-slab prior and horse-shoe prior. They are proven to be much more effective than L1/L1 +L2, and can  scale up to a large number of features/variables. They can provide uncertainty quantification, and well support non-convex loss functions. See the reference algorithms in [1][2]

[1] Ghosh, Soumya, Jiayu Yao, and Finale Doshi-Velez. "Model Selection in Bayesian Neural Networks via Horseshoe Priors." J. Mach. Learn. Res. 20.182 (2019): 1-46.
[2] Fang, S., Zhe, S., Lee, K. C., Zhang, K., & Neville, J. (2020, November). Online bayesian sparse learning with spike and slab priors. In 2020 IEEE International Conference on Data Mining (ICDM) (pp. 142-151). IEEE.

---

> ### Author Response · Authors · 2022-08-02
> **Response to review**
>
> The main contributions and recalls on Anderson acceleration are provided in the common response to the reviewers and will be added in the main text.
>
> **It seems working set and Anderson acceleration are known ideas, right? How novel is it to apply them in the coordinate descent framework? Is there any other CD algorithm that uses the two ideas?**
> Working sets, Anderson acceleration and coordinate descent are indeed known ideas. Working sets have already been combined with coordinate descent in [2] or [3]. Our point of view is to take advantage of the interplay between working sets and Anderson acceleration. As highlighted by the ablation study, Anderson acceleration happens faster when combined with working sets (see also the response to R1, "Intuition and Anderson acceleration does not help without working set strategy").
> We are not aware of previous works combining working sets, coordinate descent and (primal) Anderson extrapolation in the convex and non-convex cases.
>
> **Is the proposed score function brand-new?**
> To our knowledge, the proposed score has never being used to build working sets in the convex and non-convex cases. During our literature review, we found that this score had been used once to do Gauss-Southwell greedy coordinate descent (which is not a practical algorithm) in the convex case [4]. More details can be found in Remark 17 in Appendix C.
>
>
> **Working set size $n_k$**
> This is a usual choice in the working sets literature, which allows working sets to grow quickly, but stops growing once the coeffcient $\beta^{(t)}$ identifies the support. This geometric growth policy is backed up by theortical results in [1] Section 4.1, and the experimental results obtained in Fig. 8 and 9 of [2].
>
> **Sparse Bayesian learning**
> Sparse Bayesian learning is indeed an alternative to optimization-based models for sparse recovery.  Our use of non-convex penalties is motivated with a statistical comparisons between optimization-based estimators (Figure 1). However, our primary goal is to provide fast and flexible *algorithms / solvers* for existing optimization-based estimators, *not to provide new statistical estimators*. As we highlighted in the introduction, optimization-based estimators are still widely used in practice. Hence, we strongly believe that a statistical comparison between Bayesian and optimization-based approaches is beyond the scope of our paper.
> Nonetheless, we now point toward the proposed papers for alternative approaches to optimization-based estimators.
>
>
> [1] Ndiaye & Takeuchi (2021). Continuation Path with Linear Convergence Rate. arXiv preprint arXiv:2112.05104.
>
> [2] Massias, Gramfort & Salmon (2018). Celer: a fast solver for the lasso with dual extrapolation. ICML
>
> [3] Rakotomamonjy, A., Flamary, R., Salmon, J., & Gasso, G. (2022, May). Convergent Working Set Algorithm for Lasso with Non-Convex Sparse Regularizers. In International Conference on Artificial Intelligence and Statistics (pp. 5196-5211). PMLR.
>
> [4] Nutini, J., Schmidt, M., Laradji, I., Friedlander, M., & Koepke, H. (2015, June). Coordinate descent converges faster with the gauss-southwell rule than random selection. In International Conference on Machine Learning (pp. 1632-1641). PMLR.

---

> > ### Author Response · Authors · 2022-08-08
> > **Author feedback**
> >
> > Dear reviewer,
> > We hope you were able to read our answer and that it addresses your questions about the score, the working set size and the position with respect to Sparse Bayesian Learning. Please let us know if you have any additional feedback.

---

### Official Review · Reviewer_naRr · 2022-07-08

**Rating:** 7
**Confidence:** 3
**Soundness:** 3 good
**Presentation:** 3 good
**Contribution:** 3 good

**Summary:**

The paper proposed a new algorithm to estimate generalized linear model with separable penalties. The algorithm is based on coordinate descent, working sets and Anderson acceleration. It is implemented in a new flexible sklearn compatible package. The numerical experiments show time improvement compared to existing methods.

**Questions:**

See "Strengths and Weaknesses"

**Ethics Review Area:**

["I don’t know"]

**Limitations:**

See "Strengths and Weaknesses"

**Strengths And Weaknesses:**

The authors proposed a generic algorithm that uses coordinate descent, working sets and Anderson acceleration for generalized linear model with sparsity-induced separable penalties (which could be nonconvex). For a certain class of nonconvex penalties, they provided the theoretical analysis for the convergence, support identification, and local convergence rate of the proposed algorithm under certain regularity conditions. They implemented the algorithm in a flexible way that is compatible with scikit-learn package, which can also accommodate other penalties. They also perform extensive numerical experiments for both convex ($\ell_1$, $\ell_1-\ell_2$) and nonconvex penalties (MCP), on huge-scale real datasets synthetic. They also conduct ablation study on working sets and Anderson acceleration. The paper is well written.
The following are some questions/comments:
1. There are many assumptions and propositions in the paper, and the important propositions 10 and 13 uses multiple assumptions that may not hold in reality. Some of these assumptions like convergence to a critical point may be standard for the convergence analysis. But some of them such as Assumptions 11 and 12 might be too strong and may not hold for a general formulation, or even MCP. I think it is better to illustrate how strong these assumptions are. For example, for Proposition 13, Assumptions 1,2, and 8 and convergence towards a critical point might be standard; and piecewise quadratic holds for MCP regression; what about assumptions 9, 11 and 12 for $\hat\beta$? Because I do not think Assumption 11 and 12 will hold for any $\hat\beta$. Especially when $|\hat\beta_j|=\gamma\lambda$, Assumption 11 will not hold?
2. Nonmonotonicity of the curves: when I read the figures for the first time, I noticed this nonmonotonicity in terms of the duality gap vs time curve. After carefully reading through the paper, I found that the authors explained it in line 219 on page 7 as well as in the last paragraph of supplement. It is probably better to point out this to reader and refer to this paragraph in some of these plot captions, so as to improve readability. Furthermore, I am wondering if it is possible to fix different levels of duality gaps and plot the median time of achieve this duality gap with the package to ensure the curve’s monotonicity?
3. For the MCP plot (Figure 5), how is optimality condition in the y-axis of the bottom panel defined?

---

> ### Author Response · Authors · 2022-08-02
> **Response to review**
>
> We thank the reviewer for acknowledging the theoretical guarantees as well as the flexible and efficient implementation.
>
> **How realistic are the assumptions?**
>
> - ***Assumption 9*** (Non-degeneracy) Assumption 9 is usual (and necessary) for this kind of analysis (see for instance [2,5,6]). For the $\ell_1$ norm, if the entries of the design matrix $X$ are drawn from a i.i.d normal distribution, then Assumption 9 holds with high probability [4,7]. There exists some works attempting to relax Assumption 9 (in the convex case), however weaker results are obtained at the cost of a much complex analysis [3].
> - ***Assumption 11*** (Locally $\mathcal{C}^3$) For the datafitting term $f$, the regularity Assumption 11 is mild and holds for usual datafitting terms such as quadratic or logistic functions. For the regularizer, this assumption is stronger, and indeed implies that $\beta_j \neq \lambda \gamma$, for all $j \in \mathcal{S}$, for the MCP. However this is a standard assumption in the literature, see for instance Section 3.3 [8].
> - ***Assumption 12*** (Local strong convexity) Assumption 12 implies local strong convexity restricted to the support, which is also usual in the MCP / SCAD literature (Section 4.1 [1]) and is usual to derive linear rates of convergence (Section 3.3 [8]). For instance, for the Lasso, if the entries of the design matrix $X$ are drawn from a continuous distribution, then Asumption 12 holds with probability one (Lemma 4 [2]).
>
> All these assumptions are now thouroughly discussed in the revised paper.
>
> **Earlier reference to non-monotonicity of curves**
> This phenomenon is now documented and referenced earlier in the main next. As you suggested, we can also include median times to reach the desired suboptimality.
>
> **Optimality for the MCP figure** (Figure 5)
> The optimality condition quantifies how much the condition $0 \in \partial (f + g)(\beta^{(k)})$ is satisfied, it is defined as the distance of the negative gradient of the datafit $f$ to the subdifferential of the penalty $g$: $\mathrm{dist} \left (-\nabla f(\beta^{(k)}), \partial g(\beta^{(k)}) \right)$. This quantity is 0 for any critical point (by definition of a critical point). The value of the MCP subdifferential can be found in Section 2.2. The definition of the optimality condition has been added in the caption of Figure 5.
>
>
> [1] Breheny, P., & Huang, J. (2011). Coordinate descent algorithms for nonconvex penalized regression, with applications to biological feature selection. The annals of applied statistics, 5(1), 232.
>
> [2] Tibshirani, R. J. (2013). The lasso problem and uniqueness. Electronic Journal of statistics.
>
> [3] Fadili, J., Malick, J., & Peyré, G. (2018). Sensitivity analysis for mirror-stratifiable convex functions. SIAM Journal on Optimization.
>
> [4] Candes, E. J., & Tao, T. (2005). Decoding by linear programming. IEEE transactions on information theory.
>
> [5] Poon, C., Liang, J., & Schoenlieb, C. (2018). Local convergence properties of SAGA/Prox-SVRG and acceleration. ICML.
>
> [6] Poon, C., & Liang, J. (2019). Trajectory of alternating direction method of multipliers and adaptive acceleration. NeuRIPS
>
> [7] Rudelson, M., & Vershynin, R. (2008). On sparse reconstruction from Fourier and Gaussian measurements. Communications on Pure and Applied Mathematics.
>
> [8] Liang, J., Fadili, J., & Peyré, G. (2016). A multi-step inertial forward-backward splitting method for non-convex optimization. NeurIPS.

---

> > ### Author Response · Authors · 2022-08-08
> > **Author feedback**
> >
> > Dear reviewer,
> > We hope you were able to read our answer and that the meaning and context of our Assumptions are now clear. Please let us know if you have any additional feedback.

---

> > > ### Comment · Reviewer_naRr · 2022-08-08
> > > **Thank you for the response.**
> > >
> > > Dear authors,
> > >
> > > Thank you for the response for clarifications. The response makes sense to me. I have increased the score to 7.
> > >
> > > Best,
> > > Reviewer

---

### Official Review · Reviewer_uXjd · 2022-07-11

**Rating:** 6
**Confidence:** 2
**Soundness:** 3 good
**Presentation:** 2 fair
**Contribution:** 3 good

**Summary:**

Paper proposes a new generic algorithm for solving sparse optimization problems with very large number of variables, with convex or non-convex penalties (regularization). Their algorithm 'skglm' uses working-set strategy on the outer loop and coordinate descent with Anderson acceleration in the inner loop. Working set strategy enables scaling to millions of variables.

**Questions:**

- in the ablation study you note that the Anderson acceleration does not help unless working set strategy is used. This seems counterintuitive, isn't working set strategy quite orthogonal technique? Do you have any insight on this.
-

**Limitations:**

Yes.

**Strengths And Weaknesses:**

Strengths
- broad area of application for the proposed algorithm, surely relevant to the community
- work is placed well into the literature
- algorithm shows promising experimental results
- code is available and should be easy to use

Weaknesses
- I find the paper overly technical and hard to read. For example, would be good to include definitions of MCP and SCAD in the paper, as especially MCP is referred to many times. Similarly, it would be good to include the definition or intuition of Anderson acceleration in the paper itself.
- The intuition, story, about how how authors arrived to the solution could be clearer. As the proposed algorithm seems combination of existing methods, what is the great insight, research contribution, that leads to this solution?
- all plots are way too small and hard to understand. Consider different ways to present the results in a clearer way.


As a summary, the results look promising, even though I am not an expert of this particular area. By making the paper more readable, self-contained and intuitive, it would be more impactful. Consider limiting the other packages you compare to (you can include in the appendix) and highlighting the experiments where the algorithm particularly beats the competition,

---

> ### Author Response · Authors · 2022-08-02
> **Response to review**
>
> We thank you for acknowledging the strength of our experimental results, the availability of high quality code and the position in the literature.
>
> **Technicality**
> We have updated the submission and tried our best to ease the reading of the paper. In particular:
> - All assumptions are now thouroughy discussed.
> - MCP definition has been added as a non-convex non-smooth penalty example in the problem setting section (Section 2.1).
> - SCAD definition will be included in the main text for exposition purposes.
>
> **Intuition** and  **Anderson acceleration does not help without working set strategy** (ablation study)
> The *intuition* is that once the (generalized) support (e.g. the non-zero coefficients for Lasso) has been identified, the optimization problem is locally quadratic, and Anderson extrapolation provably provides local acceleration. Empirically, we observed that working sets strategies yield faster support identification: hence the acceleration from the Anderson acceleration happens faster when combining working sets and Anderson acceleration.
> - This phenomenon can be observed on Figure 6 for news20, $\lambda_{\max} / 100$: the Anderson extrapolation provides acceleration with and without working sets, but the acceleration happens quicker when combined with the working sets.
> - As you point out, sometimes, without the working sets strategy, Anderson extrapolation does not seem to provide speedups (e.g. new20, $\lambda_{\max} / 1000$ in Figure 6):  without the working sets strategy the support has not been idententified yet, hence the acceleration has not happend yet. In other words, if one let the algorithm without workling sets run for longer, one could see the acceleration happens, but latter.
>
>
> **Plots**
> Figures are now larger and clearer: some datasets and competitors have been removed for clarity and the modified figures appear in the revised paper. Feel free to let us know if you have more plot-related feedback.

---

> > ### Author Response · Authors · 2022-08-08
> > **Author feedback**
> >
> > Dear reviewer,
> > We hope that you had the opportunity to check the revised version we submitted, where we have improved the presentation thanks to your thoughtful remarks. We will take the rest into account within the additional page if the paper is accepted. Please let us know in the meantime if you were satisfied with our answer.

---

### Author Response · Authors · 2022-08-02
**Common response to reviewers**

We would like to thank all the reviewers for their feedback. Thanks to them the clarity of the paper has been greatly improved. Most of the proposed modifications now appear in the revised paper.
We were not able to add them all because of the page constraint (an extra page is allowed only for the camera ready), and we will add all of them if the paper is accepted. We answer common questions below, and specific ones in separate discussions.

We also would like to thank the reviewers for acknowledging the versatility of our algorithm, its practical relevance for the community, the experimental validation showing state-of-the-art results on very high dimensional problems and the availability of quality code.

**Intuition for Anderson acceleration (AA)** (Reviewers uXjd and 783Y)
Anderson acceleration has become very popular in the recent ML literature (we have added references for the relevance of this technique in ML). So far, it has mostly been applied to convex quadratic problems, while our work encompasses non-convex functions. Anderson extrapolation is a technique to accelerate the convergence of fixed-point iteration sequences:  $$\beta^{t+1} = A \beta^{t} + b   \quad , \ (1)$$ with $A \in \mathbb{R}^{p \times p}$ such that $\rho(A) < 1$. The key idea of Anderson extrapolation is to search for an approximation of the limit $\beta^*$ as a combination of the $K$ last iterates $\sum_0^{K-1} c_i \beta^{t - i}$. The coefficients $c_i$ are meant to approximate the coefficients $c^\star$ of $A$'s minimal polynomial, as it can be shown that the limit $\beta^*$ is equal to $\sum_1^p c_i^\star \beta^{t-i}$. An intuitive explanation for this is in ["Dual Extrapolation for Sparse GLMs", Massias et al., JMLR, 2020, below Proposition 6 (page 6)](https://www.jmlr.org/papers/volume21/19-587/19-587.pdf).
The two difficulties we adress are:
- Showing that the structure (1) holds after support identification (Prop 10).
- Showing that $\rho(A) < 1$, using the local strong convexity restricted to the support (Lemma 16 in Appendix).


**Research contribution** (Reviewers uXjd and 783Y)
The main first contribution is practical: we proposed a simple, fast and flexible algorithm, which surpasses previous state-of-the-art approaches. As our extensive experiments show, we believe that our contribution is of wide practical interest to the community. Let us emphasize that there does not exist fast, flexible and well-implemented algorithms for the generic problem we consider.

The main second contribution is theoretical: we show local accelerated convergence rates for some non-smooth non-convex problems, whereas this kind of results usually only hold for *quadratic problems*, see for instance [1, 2, 3]. We managed to obtain these new results by combining:  the support identification property of proximal coordinate (which is a new results) and the regularity + local strong convexity restricted to the support.

[1] Scieur, D., d'Aspremont, A., & Bach, F. (2016). Regularized nonlinear acceleration. Advances In Neural Information Processing Systems, 29.

[2] Scieur, D., Bach, F., & d'Aspremont, A. (2017). Nonlinear acceleration of stochastic algorithms. Advances in Neural Information Processing Systems, 30.

[3] Bollapragada, R., Scieur, D., & d’Aspremont, A. (2022). Nonlinear acceleration of momentum and primal-dual algorithms. Mathematical Programming, 1-38.

---

### Meta-Review · Area_Chair_X7Kz · 2022-08-26

**Recommendation:** Accept
**Confidence:** Certain

**Metareview:**

The paper proposed a fast coordinate descent algorithm for sparse linear models with \alpha-semi-convex penalties. The algorithm includes two key steps:  to introduce a score to rank the variables for obtaining a working set, and to use Anderson acceleration in the inner-loop. The theoretical analysis and the numerical experiments show the effectiveness of the proposed algorithm. The reviewers raised several concerns, which should be addressed in the final version.

**Award:**

No

---

### Decision · Program_Chairs · 2022-09-14

Accept